# Annular Surface Micromachining of Titanium Tubes Using a Magnetorheological Polishing Technique

**DOI:** 10.3390/mi11030314

**Published:** 2020-03-17

**Authors:** Wanli Song, Zhen Peng, Peifan Li, Pei Shi, Seung-Bok Choi

**Affiliations:** 1School of Mechanical Engineering and Automation, Northeastern University, Shenyang 110819, China; 1800391@stu.neu.edu.cn (Z.P.); wslipeifan@163.com (P.L.); shipeineu@163.com (P.S.); 2Key Laboratory of Vibration and Control of Aero-Propulsion Systems Ministry of Education of China, Northeastern University, Shenyang 110819, China; 3Department of Mechanical Engineering, Inha University, Incheon 22212, Korea

**Keywords:** magnetorheological (MR) fluid, MR polishing, internal surface, titanium-alloy tube, surface roughness

## Abstract

In this study, a novel magnetorheological (MR) polishing device under a compound magnetic field is designed to achieve microlevel polishing of the titanium tubes. The polishing process is realized by combining the rotation motion of the tube and the reciprocating linear motion of the polishing head. Two types of excitation equipment for generating an appropriate compound magnetic field are outlined. A series of experiments are conducted to systematically investigate the effect of compound magnetic field strength, rotation speed, and type and concentration of abrasive particles on the polishing performance delivered by the designed device. The experiments were carried out through controlling variables. Before and after the experiment, the surface roughness in the polished area of the workpiece is measured, and the influence of the independent variable on the polishing effect is judged by a changing rule of surface roughness so as to obtain a better parameter about compound magnetic field strength, concentration of abrasive particles, etc. It is shown from experimental results that diamond abrasive particles are appropriate for fine finishing the internal surface of the titanium-alloy tube. It is also identified that the polishing performance is excellent at high magnetic field strength, fast rotation speed, and high abrasive-particle concentration.

## 1. Introduction

Titanium-alloy tubes have been increasingly applied in aerospace, military, chemical, and other fields to transport corrosive gas or liquid because they feature low density, high specific strength, and superior corrosion resistance [1]. However, surface defects on the inner surface of titanium tubes caused by the machining process have a significant influence on their mechanical properties and lifespan. Thus, it is necessary to polish the inner surface of titanium tubes to reduce surface roughness and eliminate surface defects. However, owing to the low thermal conductivity and high chemical reactivity of cutting-tool materials, it is difficult to polish the inner surface of titanium-alloy tubes [2,3]; the traditional mechanical polishing technologies cannot meet the current processing-efficiency and accuracy demands, and they are labor intensive. In addition, the tubular shape increases the difficulty of processing. Therefore, innovative polishing methods are being developed to address these problems encountered when polishing the inner surface of titanium-alloy tubes.

To date, numerous polishing processes have been developed for titanium alloys to achieve a smooth surface [4]. Based on traditional chemical–mechanical polishing (CMP), an environment-friendly slurry consisting of silica, hydrogen peroxide, malic acid, and deionized water was prepared in a previous study to polish titanium alloys. The resulting surface roughness of the polished specimens was low, and the corrosion potential was congruent with the quality of the polished surface [5]. Electrical discharge machining (EDM) is a non-conventional process that is carried out using the negative polarity of different materials to polish titanium alloys. Results of a previously conducted study indicate that using a copper electrode yields the lowest surface roughness, whereas using a graphite electrode yields the highest surface roughness [6]. Laser polishing has also been used to finish titanium alloys; it effectively improves the surface quality without damaging the substrate [7]. In one experiment, the laser polishing reduced the initial surface roughness of the workpiece from more than Ra 5 μm to less than Ra 1μm, and the surface microhardness and wear resistance were significantly enhanced. Titanium alloys and other types of hard alloys can also be machined by electrochemical honing (ECH), which combines an electrochemical machining process and a conventional honing process to accurately control the finishing forces and achieve precise polishing [8]. Although the aforementioned polishing processes can polish titanium alloys, accurate control of the finishing forces is difficult to achieve for most of them.

Magnetorheological (MR) polishing achieves accurate control of polishing forces based on the rheological properties of an MR polishing fluid under a magnetic field [9,10,11,12]. During the MR polishing process, an MR polishing fluid is transformed into a stiffened fluid ribbon that acts as the polishing tool. The stiffness and shape of the MR polishing fluid ribbon can be altered by increasing the magnetic field strength [13,14]. In addition, the types of abrasive particles used in the MR polishing fluid have a significant impact on the polishing performance for different materials. Cerium oxide abrasive particles may be added into the MR polishing fluid to polish a silicon sample or other related silicate glasses [15], whereas SiC or diamond abrasive particles are typically used to polish metal materials [16,17,18] In recent years, several methods of polishing the internal surfaces of cylindrical workpieces using MR polishing have been reported. For example, a magnetorheological abrasive flow finishing (MRAFF) process was developed to polish workpieces with complex internal geometries [19,20]. The MR polishing fluid carried out reciprocating linear motion by using a hydraulically powered device. Stainless steel flat workpieces were polished such that the initial surface roughness decreased from Ra 0.47 to Ra 0.34 μm when a magnetic field strength of 0.574 T was applied. Thereafter, the rotational motion of excitation equipment was added into the MRAFF process, and a rotational–magnetorheological abrasive flow finishing (R-MRAFF) process was proposed to improve the polishing quality [21,22,23]. A uniformly smooth mirror was obtained in which helical abrasive particle trajectories were formed by rotation and reciprocating motion. A novel MR polishing process based on magnetorheological abrasive honing (MRAH) was introduced to finish the internal surface of a ferromagnetic cylindrical workpiece [24,25]. The electromagnet was driven by a rotating shaft, resulting in rotational motion and reciprocating linear motion, which made the abrasive particles translate and rotate, respectively, passing through the workpiece surface as in the case of the honing process.

However, no experiment has been conducted to investigate the polishing of the internal surface of titanium-alloy tubes using existing MR polishing processes. Typically, an electromagnet or permanent magnet is used as the excitation equipment in MR polishing devices, and an electromagnet is used in MRAH. However, a certain number of coils are required to guarantee sufficient magnetic field strength, which makes it difficult to polish the inner surfaces of cylindrical workpieces. Influenced by the heat generated by electromagnets, the polishing performance of the MR polishing fluid will not achieve the desired result. On the contrary, no generation of heat and occupation of a small space are the excellent advantages of permanent magnets compared with electromagnets. A permanent magnetic yoke generates the magnetic field [26], the strength of which in the polishing area can only be adjusted roughly by changing the excitation gap or working gap. Therefore, the finishing forces cannot be accurately controlled in this process. The combination of a permanent magnet and an electromagnet is expected to achieve accurate control of the finishing forces and a reduction of both the occupied space and the heat generated by the coils. Additionally, an electromagnet can adjust the magnetic field strength based on the magnetic field strength provided by the permanent magnet. In this study, a novel MR polishing process is developed to finish the internal surface of a titanium-alloy tube. Two types of excitation equipment are designed to generate a compound magnetic field. The effects of compound magnetic field strength, rotation speed, and types and concentrations of abrasive particles on the polishing performance are systematically investigated.

## 2. Design of the MR Polishing Apparatus

### 2.1. MR Polishing Apparatus

The MR polishing experiment apparatus has been designed for polishing the inner surface of titanium tube, which is used for aviation pipelines. The basic requirement of the proposed apparatus includes the following points:The structure is compact and easy to install.Recycling of MR polishing fluid is realized to maximize the material utilization.The polished tube is allowed to rotate at high speed.Reciprocating linear motion of polishing head is an indispensable function, which should be considered into the design of MR polishing apparatus.The movement speed of MR polishing device is controllable.

The whole apparatus is as given in Figure 1. The tube is axially clamped by fixtures on two sides and left fixture is connected with an adjustable speed motor by coupling to drive the tube to rotate at high speed. For convenient disassembly of a workpiece, the coupling connecting to the right fixture is installed on the platform where a spacing sliding block is mounted. The guide rail matched with the spacing sliding block is fixed with the body frame so that the right fixture can be driven away from or approaching to the workpiece by moving the platform. The polishing head is merged with one end of the polishing shaft and put inside the tube. The space between the tube and the polishing head is saturated with MR polishing fluid, which gets stiffened under the action of excitation equipment. The neutral of polishing shaft can be controlled by the X–Z working table. The excitation equipment and the X–Z working table are bound to the ball screw driven by the stepping motor to do synchronous reciprocating linear motion and the speed is controlled by the single-chip computer. The carrier tank is installed to connect the two ends of tube to recycle the MR polishing fluid, which is stored in the carrier tank and delivered during the polishing process by the peristaltic pump.

### 2.2. Polishing Mechanism

The polishing mechanism is shown in Figure 2. Excitation equipment is installed on the outside of the tube to let magnetic induction lines pass through the interior of tube. Under the action of an external magnetic field, magnetic particles are magnetized and aggregated into chains along the direction of the magnetic field, which is manifested in the macroscopic scale as stiffening of the MR polishing fluid. Nonmagnetic abrasive particles are embedded between the magnetic-particle chains [27,28] and connected with the inner surface of the tube to remove the material. When the magnetic field intensity increases, the connection between the magnetic chains also gets stronger leading to bigger yield shear stress of the MR polishing fluid, which means the rough inner surface is easier to polish. In such a situation, two-body abrasion mainly happens to the polishing abrasive particles and the tube surface and the polishing efficiency is relatively high. However, under a weaker magnetic field, the yield shear stress of MR polishing fluid reduces and insufficient shearing force is provided to polish. Three-body abrasion mainly occurs, which suggests low polishing efficiency. During the polishing process, the flexible polishing ribbon forms because of the solidification of the MRF polishing fluid under the external magnetic field and then moves in a synchronous reciprocating linear motion with the polishing head and the excitation equipment to cover the whole area to be polished. Meanwhile the tube keeps rotating at an appropriate high speed to shear the material. This polishing process is similar to the honing process and a spiral path is formed by the polishing particles on the inner surface of the tube. In this paper, the circulation system of the MR polishing fluid was designed as demonstrated in Figure 3. The arrows indicate the flow direction of the MR polishing fluid, which is pumped into the interior of the polishing head through the hole in the polishing shaft. Rectangular holes are uniformly distributed along the circumferential direction of the polishing head, allowing the MR polishing fluid to flow into the polishing area between the internal surface of the tube and the polishing head. Under a magnetic field, owing to its continuous delivery, the MR polishing fluid with small apparent viscosity and yield stress runs from the polishing area to both ends of the tube; the MR polishing fluid is collected from both ends of the tube, and thus it continues participating in the MR polishing process. In this way, the proposed MR polishing device completes its task of polishing the inner surface of titanium tubes and recycles the MR polishing fluid. It is noted that the MR polishing fluid will be circulated only when polishing the same one titanium alloy tube. Since the abrasive particles are damaged after multiple cycles, the MR polishing fluid will not participate in the polishing of the next tube. If another workpiece needs to be polished, the MR polishing fluid will be replaced with a new one. 

The forces applied to a single abrasive particle during the polishing process are shown in Figure 4. While the abrasive particle is working, it will be pressed into the workpiece surface under the pressure force *F_n_* forming an indentation with the depth of *d_p_* (refer to Equation (3)), and pushed by shear force *F_s_* to polish. The abrasive particle will be accordingly obstructed by the obstructing shear action *R_s_* (see Equation (1)) generated by the workpiece surface. So, *R_s_* and *F_s_* are a pair of opposite forces. Different numerical values of *R_s_* and *F_s_* will lead to different effects. *F_s_* is the shear force applied by the magnetic particle chains. When *F_s_* > *R_s_*, the abrasive particle held by the magnetic chains can remove the surface material; however, when *F_s_* < *R_s_*, the magnetic chains may not provide enough shear force to remove the surface material, leading the abrasive particle to roll on the workpiece surface, which makes the polishing behavior inefficient or even prevents the polishing behavior from happening. Therefore, by comparing *F_s_* and *R_s_*, whether the polishing action can be carried out smoothly can be known. Otherwise, moderate adjustments should be conducted properly.

The obstructing shear action is given as follows:(1)$$R_{s}=A_{p}\sigma$$
where, *σ* is the yield strength of the machined material with a value of 895 MPa, the cross-sectional area of indentation *A_p_* is expressed as:(2)$$A_{p}=\frac{d_{a}^{2}}{4}\sin^{-1}\frac{2\sqrt{d_{p}(d_{a}-d_{p})}}{d_{a}}-\sqrt{d_{p}(d_{a}-d_{p})}(\frac{d_{a}}{2}-d_{p})$$
and *d_p_* the depth of indentation is calculated by the following equation:(3)$$d_{p}=\frac{d_{a}}{2}-\frac{1}{2}\sqrt{d_{a}^{2}-d_{i}^{2}}$$

### 2.3. Design and Validation of the Excitation Equipment

As shown in Figure 5a,b, two types of excitation equipment were designed to analyze the polishing performance under low and high compound magnetic field strengths, respectively. In layout *α* the permanent magnets were mounted on the end face of an iron core (Figure 5a). In layout *β*, the permanent magnets were mounted at the upper and lower sides of the iron core (Figure 5b). The number of coil turns was 500. The electromagnet can adjust the magnetic field strength based on the magnetic field strength provided by the permanent magnet, which can reduce the number of coil turns to decrease coil heat and save assembly space.

The finite element software ANSYS Maxwell was used to simulate the static magnetic field based on the two types of excitation equipment. The distribution of magnetic flux density inside the tube when the excitation current was 2 A is shown for layout *α* and layout *β* in Figure 6a,b, respectively. The magnetic flux density for layout *α* was approximately 0.5 T; however, the magnetic flux density for layout *β* was approximately 0.2 T. It was observed that the magnetic flux density inside the polishing head was close to 0 T in both types of excitation equipment, which implies that the flow of the MR polishing liquid inside the polishing head was not affected by the magnetic field. This can be attributed to the fact that the polishing head was made using a ferromagnetic material, which can shield the magnetic field.

For further comparison, point A in Figure 6a was elected as a reference point. Figure 7 illustrates the magnetic flux density of point A in layout *α* and layout *β* under different excitation currents. The variation in the magnetic flux density with respect to the excitation current was the same for both types of excitation equipment. When the excitation current was positive, the compound magnetic field strength increased with an increase in excitation current. However, when the excitation current is negative, the compound magnetic field strength is expected to decrease with an increase in excitation current. This can be explained by the fact that the magnetic field directions of the electromagnetic and the permanent magnet are the same under a positive excitation current but opposite under a negative excitation current. Apparently, when the same excitation current is applied, the magnetic flux density of point A in layout *α* is much larger than that in layout *β*, whereas the gradient of magnetic flux density varying with the excitation current in layout *α* is smaller than that in layout *β*. This phenomenon is a result of the dominance of magnetic flux density generated by the permanent magnet in layout *α*, whereas in layout *β*, the magnetic flux density generated by the electromagnet is dominant.

To verify the correctness of the magnetic field simulation, an HT201 tesla meter was used to measure the actual magnetic flux density. As shown in Figure 7, the simulation results closely coincide with the experimental results. Hence, layout *β* was selected to generate a low magnetic flux density less than 0.2 T, and layout *α* was selected to generate a high magnetic flux density of approximately 0.45 T.

## 3. Experimentation

### 3.1. Preparation of the MR Polishing Fluid

The MR polishing fluid was prepared by dispersing microsized magnetic particles and nonmagnetic abrasive particles into a water-based or oil-based medium. The reversible rheological properties of the MR polishing fluid under a magnetic field is substantially attributed to the magnetic particles, which carbonyl iron particles (CIPs) with high permeability and low hysteresis was selected as. However, the nonmagnetic abrasive particles actually played a leading role in removing the surface material. Diamond abrasive particles or SiC abrasive particles served as the nonmagnetic abrasive particles in the titanium-alloy tube polishing process; their polishing performance is discussed in the following section. To maintain the low apparent viscosity of the MR polishing fluid under a zero magnetic field, deionized water was selected as the base medium. Glycerol was added to the MR polishing fluid as a stabilizer to prevent the sedimentation of CIPs [29]. The specific parameter of the MR polishing fluid is given in Table 1.

### 3.2. Polishing Settings

As shown in Figure 8, the experiments were performed with the titanium-alloy tubes (22 mm outer diameter, 18 mm inner diameter, 100 mm long). In consideration of the deviation of the initial surface roughness, specimens with similar initial surface roughness were selected for the same group of MR polishing experiments. Four groups of experiments were conducted to investigate the effect of compound magnetic field strength, rotation speed, type of abrasive particles, and concentration of abrasive particles on the polishing performance, with specific polishing conditions summarized in Table 2 and Table 3. In order to distinguish the polished areas and unpolished areas more clearly in the same tube, partial polishing of the internal surface of the tube was necessary. Hence, the reciprocating stroke was set as 10 mm to polish a specified section of tube. The reciprocating linear speed was set as 8 cycles per minute and the total number of cycles was 675.

The distance between the inner wall of titanium alloy tube and the polishing head directly affects the magnetic field strength within the polishing area. To find the appreciate polishing gap, the magnetic field simulation test were conducted with the polishing gap of 0.5 mm, 1.0 mm, 1.5 mm, and 2.0 mm respectively. The influencing rule of polishing gap on the magnetic field strength in the polishing area is shown in Figure 9: The magnetic field strength in the polishing area is inversely proportional to the size of the polishing gap. When the polishing gap increases from 0.5 to 2.0 mm, the maximum magnetic induction intensity in the polishing area decreases from 0.6 to 0.3 T. Although an appropriate magnetic field strength can make the polishing efficiency reach the expectation, the polishing gap of 1.5 mm between the polishing head and the internal surface of the tube would be applied considering the influence of polishing gap on magnetic field, size of abrasive chain and MR fluid flow.

### 3.3. Measurement of Surface Roughness and Removed Mass

The variation of surface roughness and material mass removed compared with the surface roughness and material mass of the initial tube was utilized to assess the polishing performance. The Olympus three-dimensional (3D) laser microscope with the highest resolution of 10 nm was used to measure the arithmetical mean deviation of the profile Ra. It can obtain the surface information of the test piece by a non-contact way, without damage to the workpiece surface. It also can catch the tiny contour and is very convenient to obtain the surface image. When the originally received titanium-alloy tubes cannot meet the request of initial surface roughness, preprocessing the workpiece needs to be performed to reach the range of initial surface roughness determined in the experiments. After being polished, the titanium-alloy tube needs to be cut in half or smaller pieces, which is determined by the measuring instrument, and then use the Olympus 3D laser microscope to measure. In order to minimize the man-made operation error and machine error, three points at different locations were selected as the sampling points to measure the surface roughness before and after polishing. Three repeated measurements at each point were taken to obtain the average value of these three data as the experimental data. If the collected data fluctuates greatly, then that datum will be regarded as invalid and another measurement would be conducted to supplement test data. Electronic analytical balance was used to measure the specimen mass. Similarly, the mass of each specimen was measured three times before and after the experiments to get the average values of decreased mass as experimental data.

## 4. Results and Discussion

### 4.1. Effect of Compound Magnetic Field Strength on Surface Roughness

A series of titanium-alloy tubes with initial surface roughness in the range of Ra 1.06–1.1 µm (average surface roughness is Ra 1.09 μm) were used to investigate the effect of compound magnetic field strength on polishing performance, according to the Test No.1 in Table 3. The variation of surface roughness and material mass with a magnetic field strength is shown in Figure 9a,b, respectively. From the graph we can know that when the magnetic flux density was 0 T, 0.07 T, 0.16 T, and 0.45 T, the final surface roughness of the inner surface of the tube was Ra 1.082 μm, Ra 0.904 μm, Ra 0.793 μm, and Ra 0.633 μm respectively, and the material removal amount was 0 mg, 0.3 mg, 0.8 mg, and 1.6 mg, respectively.

Under a zero magnetic field, the surface roughness and removed material mass changed little, because in this case it was the hydrodynamic pressure that mainly acted on the abrasive particles, which were randomly distributed in the base fluid. Due to the low apparent viscosity of the base fluid, the motion of abrasive particles was almost unrestricted. Hence, the abrasive particles may easily roll on the internal surface of the tube while cutting the material, which prevents the tube from being polished. With an external magnetic field was applied, the CIPs developed a chain structure in the direction of the magnetic field. Then, the CIP chains and hydrodynamic pressure together pressed the abrasive particles toward the internal surface of the tube. Therefore, the movement of abrasive particles was obstructed by the CIP chains, which were influenced by the magnetic field strength. In Figure 10, the apparent variations in surface roughness and material mass before and after polishing were observed. The internal surface quality improved with increasing magnetic field strength. With increasing applied magnetic field strength, the magnetic forces acting on the abrasive particles also increased, and the movement of the abrasive particles was more seriously restricted by CIP chains. The initial surface roughness decreased from Ra 1.1 to Ra 0.633 μm after 675 cycles, and the removed material mass was 0.0016 g, shown in Figure 10a. The largest variation of surface roughness and material mass was attained at a magnetic field strength of 0.45 T, shown in Figure 10b. The polished titanium-alloy tube under a magnetic flux density of 0.45 T is shown in Figure 11, where the polished area appeared much brighter than the unpolished area. Figure 12 shows the hyper focal microstructure images taken of the polished surface with ultra-depth of the field microscope. The original pits on the tube surface were worn away, obtaining the desired polishing effectiveness.

### 4.2. Effect of Rotation Speed on Surface Roughness

To investigate the effect of rotation speed of titanium-alloy tubes on polishing performance, tubes with initial surface roughness in the range of Ra 0.684–0.784 μm were chosen in this group of experiments, according to Test No.2 in Table 3. The rotation speed was set from 420 to 840 rpm in increments of 140 rpm, and the results after 675 cycles are depicted in Figure 13a,b, respectively. Before reaching a rotation speed of 700 rpm, the abrasive particles cut the surface material at a higher frequency, and thus the variation of surface roughness and material mass increased with increasing rotation speed. However, when the rotation speed exceeded 700 rpm, a decreasing tendency emerged, as shown in Figure 13b. This was because the yield stress of the stiffened MR polishing fluid decreased and the chain structures were destroyed to different degrees at high rotation speeds [30], and so the restriction of abrasive particle movement by the CIP chains was weakened. Although abrasive particles cut the material more frequently, the probability of abrasive particles rolling on the internal surface of the tube also increased due to the decreased yield stress. As a consequence, more attention needs to be given to the critical rotation speed of the MR polishing fluid to achieve the best performance. The largest variation of surface roughness and material mass in these experiments appeared when the rotation speed reached 700 rpm, as shown in Figure 12. At this rotation speed, the initial surface roughness decreased from Ra 0.769 to Ra 0.401 μm and the removed material mass was 0.0016 g.

### 4.3. Effect of the Type of Abrasive Particle on Surface Roughness

Experiments to assess the role of the abrasive particles in MR polishing and to select the most appropriate type of abrasive particles for polishing titanium alloys were carried out, according to Test No.3 in Table 3. The size of abrasive particles used here in this section is claimed in Table 1 (Diamond Φ20 µm, SiC Φ25 µm). Titanium-alloy tubes with initial surface roughness in the range of Ra 0.721–0.772 μm were selected as specimens and different kinds of MR polishing fluids were prepared. The experimental results are exhibited in Figure 14. When abrasive particles were not added to the MR polishing fluid, only subtle variations of surface roughness and material mass were noticed. The initial surface roughness decreased from Ra 0.749 to Ra 0.689 μm after 675 cycles, and the removed material mass was only 0.0004 g. Due to the use of a non-abrasive particle in the MR polishing, abrasion action only occurred between the CIPs and the machined material surface, which explains the low polishing efficiency. Adding abrasive particles to the MR polishing fluid obviously promoted variations in the surface roughness and material mass. Compared with SiC, diamond abrasive particle performed much better. The surface roughness and material mass were reduced by Ra 0.299 μm and 0.0013 g, respectively, with diamond but only by Ra 0.178 μm and 0.0011 g, respectively, with SiC. Therefore, diamond abrasive particles with high hardness are much more appropriate for finishing titanium-alloy tubes in terms of polishing performance.

### 4.4. Effect of the Concentration of Abrasive Particles on Surface Roughness

Experiments were carried out to investigate the effect of concentration of abrasive particles on polishing performance, according to Test No.4 in Table 3. Titanium-alloy tubes with initial surface roughness in the range of Ra 0.609–0.651 μm were used. The concentration of diamond abrasive particles was varied from 5% to 20% in increments of 5% while the CIP concentration remained constant at 30%. As shown in Figure 15a,b, the reduction of surface roughness and removed material mass, respectively, tended to increase with increasing concentration of abrasive particles. When the concentration of abrasive particles was 5%, the polishing effect was not significant. However, when it reached 10%, the polishing effect was clearly promoted. This is because the higher the concentration of abrasive particles, the greater the number of the abrasive particles in contact with the machined alloy. Figure 16 shows the distribution of CIPs and nonmagnetic abrasive particles in stiffened MR polishing fluids with different concentrations of abrasive particles. In Figure 16a, no abrasive particles were added to the MR polishing fluid, and CIPs assembled into chain structures along the magnetic field direction. However, when low concentrations of abrasive particles were added to the MR polishing fluid, the abrasive particles showed a tendency to disrupt the CIP chain structure. Since the magnetic field strength near the internal surface of tube was higher compared with other polishing areas, most CIPs were moved by a magnetic force to approach the internal surface of the tube and accumulate together. A tiny number of nonmagnetic abrasive particles got packed into the CIP chains and accessed the internal surface of the tube, leading to the low polishing efficiency. With the increasing concentration of abrasive particles, more abrasive particles were embedded between the CIP chains, leading to the increasing number of defective CIPs chains. Hence, at high concentrations of abrasive particles, a higher number of abrasive particles are likely to come into contact with the internal surface of tube. It was confirmed that the polishing of the internal surface improved remarkably with high concentrations of abrasive particles added to the MR polishing fluid, as shown in Figure 16b,c. Therefore, improving the quantity of abrasive particles in contact with the internal surface of the tube is essential to promoting the polishing efficiency. Considering the polishing effect and making the best of materials, etc., 10% will be considered as the appropriate concentration of abrasive particles.

## 5. Conclusions

A new MR polishing method was proposed, and its effectiveness was proven through the polishing of the internal surface of a titanium-alloy tube. Two types of excitation equipment were chosen to generate the compound magnetic field required to realize the polishing process, and one of the two was determined after analysis to carry on with the subsequent work. A series of experiments were conducted through a controlling variable to investigate the effect of a compound magnetic field strength, rotation speed, and type and concentration of abrasive particles on polishing performance, which is judged using the surface roughness Ra. The results investigated in this study can be summarized as follows:A novel MR polishing device under compound magnetic field was designed to achieve microlevel MR polishing of the titanium alloy tubes. The polishing process is realized by combining the rotation motion of the tube and the reciprocating linear motion of the polishing head. Before and after the experiment, the surface roughness Ra within polished area of workpiece was measured, and through the changing rule of surface roughness, the influence of the independent variable on the polishing effect was judged.Under a zero magnetic field, little variation of surface roughness and material mass was observed. Under an external magnetic field, the polished surface exhibited a significant improvement, particularly with higher magnetic field strengths. The initial surface roughness Ra decreased to 57.5% under a magnetic flux density of 0.45 T.Prior to the rotation speed reaching 700 rpm, the reduction of surface roughness and material mass tended to increase with an increase in rotation speed. However, high rotation speeds exceeding 700 rpm damaged the CIP chain structure and worsened the polishing performance, leading to a slowing down of the downward trend of final surface roughness Ra.When no abrasive particles were used in MR polishing, the abrasion caused by CIPs was far from the desired polishing effect. When SiC or diamond were added as abrasive particles to the MR polishing fluid, the polishing efficiency improved. Compared with SiC particles, diamond particles performed better in MR polishing: The surface roughness reduced by Ra 0.299 μm with diamond but only by Ra 0.178 μm with SiC.Low concentrations of abrasive particles led to a low polishing efficiency owing to the small number of abrasive particles in contact with the internal surface of the tube. Higher concentrations resulted in a greater number of abrasive particles available for the polishing process. From the analysis of surface roughness Ra obtained by polishing the workpiece with different concentration of polishing particles, 10% is the appropriate concentration of abrasive particles to make the best of materials and get a good polishing performance.

The optimal conditions of the rotation speed, size and shape of the polishing particles, and concentration of abrasive particles need to be explored in the future to achieve the best polishing performance.

## Figures and Tables

**Figure 1 micromachines-11-00314-f001:**
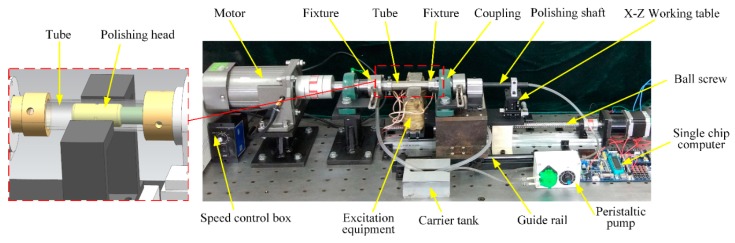
Experimental apparatus of magnetorheological (MR) polishing.

**Figure 2 micromachines-11-00314-f002:**
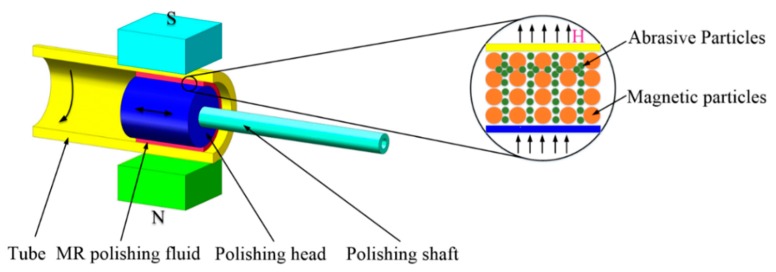
Schematic diagram of the MR polishing process.

**Figure 3 micromachines-11-00314-f003:**
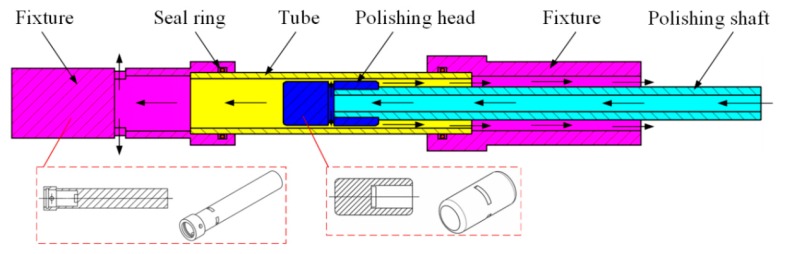
Flow direction of the MR polishing fluid.

**Figure 4 micromachines-11-00314-f004:**
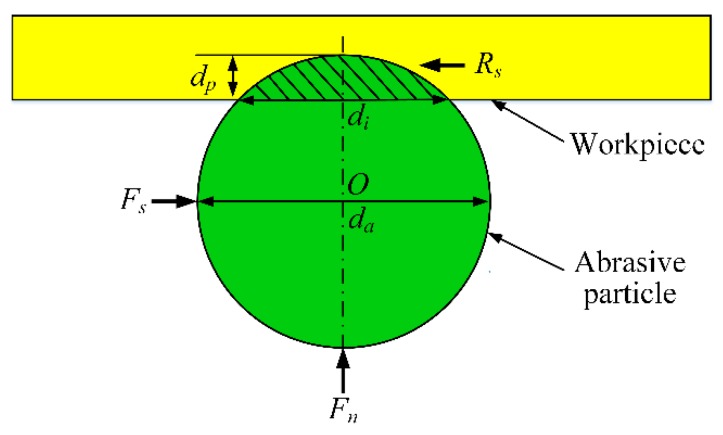
Finishing forces acting on a single abrasive particle.

**Figure 5 micromachines-11-00314-f005:**
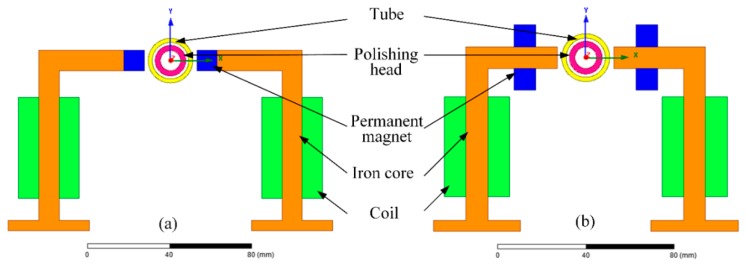
Configurations of excitation equipment: (**a**) layout *α* and (**b**) layout *β*.

**Figure 6 micromachines-11-00314-f006:**
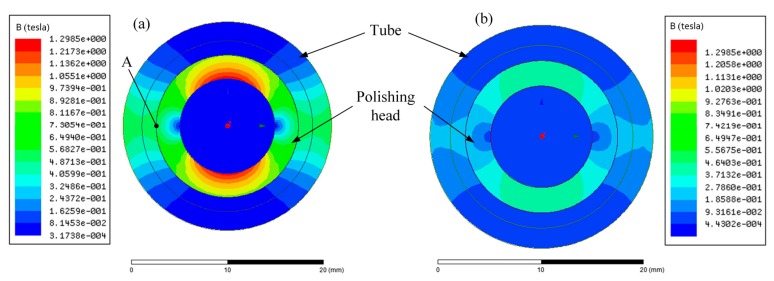
Magnetic flux density distribution inside the tube under an excitation current of 2 A: (**a**) layout *α* and (**b**) layout *β*.

**Figure 7 micromachines-11-00314-f007:**
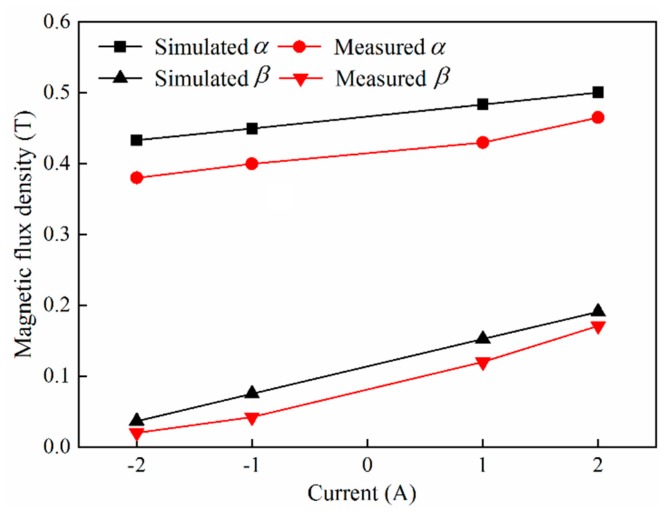
Magnetic flux density of point A under different excitation currents.

**Figure 8 micromachines-11-00314-f008:**
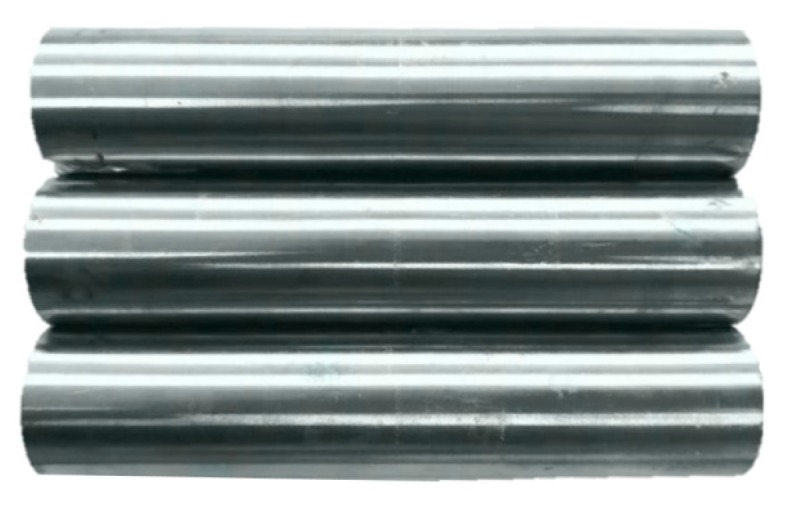
Titanium-alloy tubes.

**Figure 9 micromachines-11-00314-f009:**
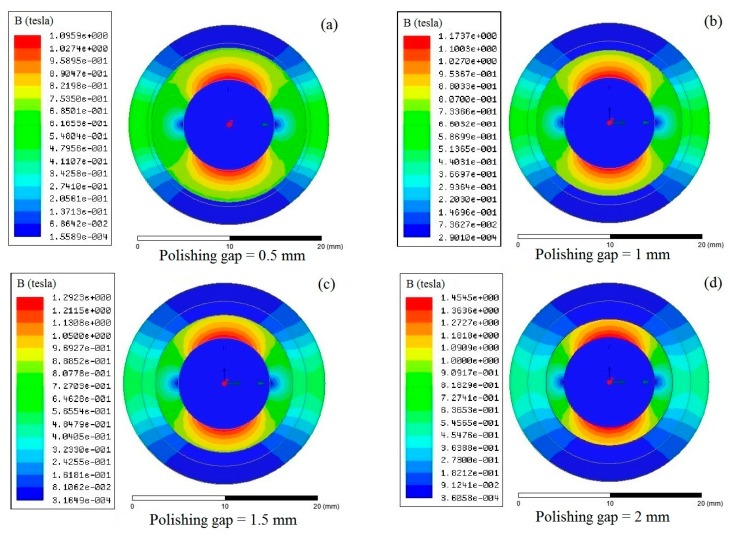
Distribution of magnetic induction intensity with different polishing gaps.

**Figure 10 micromachines-11-00314-f010:**
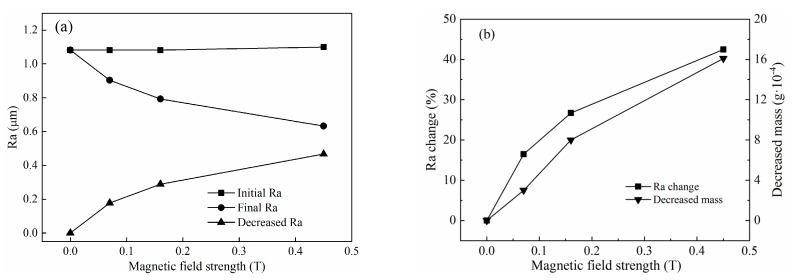
Effects of magnetic flux density on polishing performance: (**a**) surface roughness and (**b**) percentage change in surface roughness and decreased material mass.

**Figure 11 micromachines-11-00314-f011:**
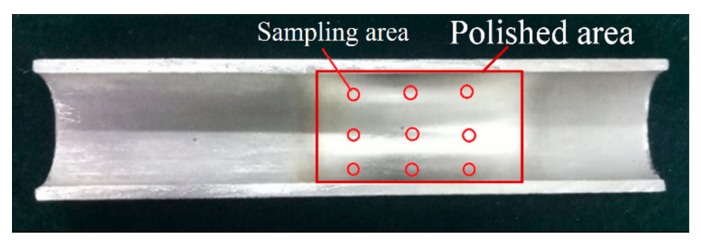
The polished titanium-alloy tube under a magnetic flux density of 0.45 T.

**Figure 12 micromachines-11-00314-f012:**
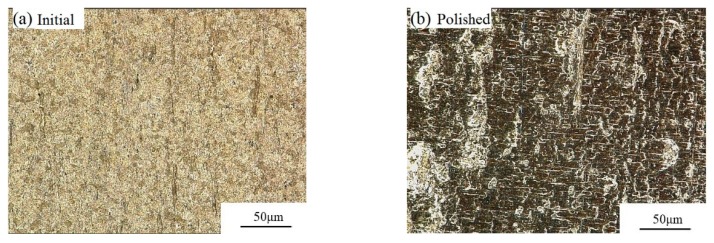
Optical microscope images of the polished surface at 1500 × magnification: (**a**) before and (**b**) after being polished.

**Figure 13 micromachines-11-00314-f013:**
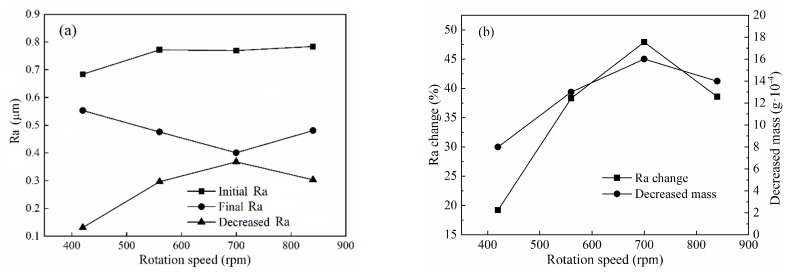
Effects of rotation speed on polishing performance: (**a**) surface roughness and (**b**) percentage change in surface roughness and decreased material mass.

**Figure 14 micromachines-11-00314-f014:**
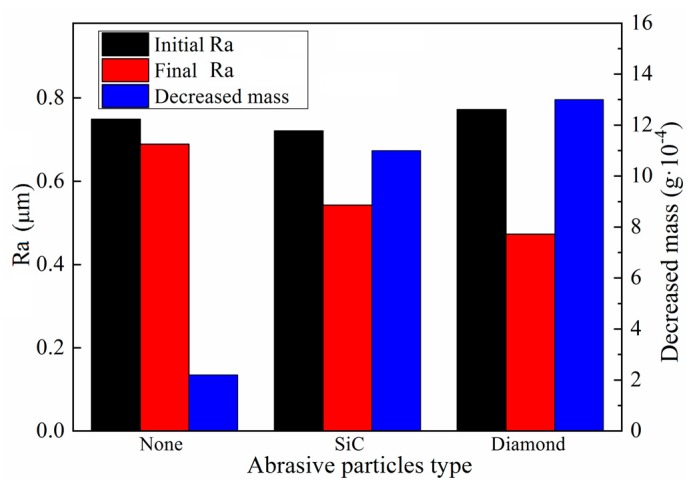
Effects of different types of abrasive particles on polishing performance.

**Figure 15 micromachines-11-00314-f015:**
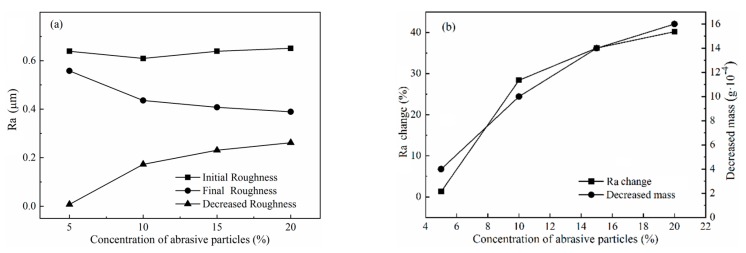
Effects of the volume of abrasive particles on polishing performance: (**a**) surface roughness and (**b**) percentage change in surface roughness and decreased material mass.

**Figure 16 micromachines-11-00314-f016:**
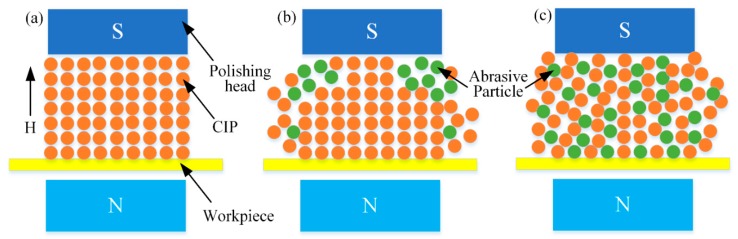
Distribution of particles in the MR polishing fluid under a magnetic field: (**a**) no abrasive particles, (**b**) low concentration of abrasive particles, and (**c**) high concentration of abrasive particles.

**Table 1 micromachines-11-00314-t001:** Composition of the MR polishing fluid.

Constituents of MR Polishing Fluid	Concentration (%)	Size
CIPs	30	Φ18 μm
Abrasive particles (SiC)	0–20	Φ 25 μm
Abrasive particles (Diamond)	0–20	Φ 20 μm
Glycerol	8	
Deionized water	42–62	

**Table 2 micromachines-11-00314-t002:** Experimental conditions.

Parameter	Values
Titanium-alloy tubes	22 mm × 18 mm × 100 m
Polishing gap	1.5 mm
Reciprocating stroke	10 mm
Reciprocating linear speed	8 cycles per minute
Reciprocating linear cycles	675
Feeding speed	3 mm/s

**Table 3 micromachines-11-00314-t003:** Setups of parametric experiments.

Test No.	Magnetic Flux Density (T)	Rotation Speed (rpm)	Type of Abrasive Particles	Concentration of Abrasive Particles (%)
1	0	560	diamond	10
	0.07	560	diamond	10
	0.16	560	diamond	10
	0.45	560	diamond	10
2	0.45	420	diamond	10
	0.45	560	diamond	10
	0.45	700	diamond	10
	0.45	840	diamond	10
3	0.45	560	no abrasive particles	10
	0.45	560	SiC	10
	0.45	560	diamond	10
4	0.45	560	diamond	5
	0.45	560	diamond	10
	0.45	560	diamond	15
	0.45	560	diamond	20

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
