# Peer review of "Annular Surface Micromachining of Titanium Tubes Using a Magnetorheological Polishing Technique"

_micromachines, 2020, doi:10.3390/mi11030314_

Round 1

Reviewer 1 Report

In this paper authors have presented MR polishing device and its performance while polishing Ti-alloy tube. They have used two types of abrasive particles and tested effect of rpm, abrasive concentration on material removal and surface finish. There has been multiple studies of similar nature in the literature hence it is necessary that authors should highlight how their study is original and advances the knowledge of the field. Writing needs to be checked for grammatical error as few sentences end with prepositions.

Regarding the setup, authors should compare the study with internal polishing tools presented by (Yamaguchi H, Kang J, Hashimoto F. CIRP annals, 2011, 60(1): 339-342 ; Zhang J, Hu J, Wang H, et al. Precision Engineering, 2018, 54: 222-232; and Yun H, Han B, Chen Y, et al. The International Journal of Advanced Manufacturing Technology, 2016, 85(1-4): 727-734.).

The gap between the polishing head and workpiece inner surface is very important, since it will affect the magnetic field, size of abrasive chain and MR fluid flow. However, its effect is not addressed in detail in the study. (Line 244: The polishing gap between the polishing head and the 244 internal surface of tube was 1.5mm)

It is claimed that MR fluid and abrasive media can be recirculated however it is well known that the abrasive particles are damaged after few cycles. Analysis of the particles before and after the polishing must be added.

How many measurements were performed for surface roughness? If the average values are presented it should be given with the error bars.

In Fig 10, the polishing looks uneven so it is important that the authors provide the surface roughness profile in the paper instead of just average values.

Fig 11 is not SEM image. Just optical microscope images.  

Eqs 1-3 are added with limited use in the discussion. How these are related to the current process must be discussed in details.  

Reviewer 2 Report

Congratulations to the authors for committing a valuable article that will be ready for publication after applying the following corrections:

  1. Line 52: The authors wrote: „surface roughness of the workpiece from more than 5mm to less than 1mm” Please specify what parameter the authors used to evaluate surface roughness, maybe Ra or Sa? A similar error appears throughout the article, please analyze all the work in this regard. In particular, reference should be made to the parameters for assessing surface roughness in Conclusions.
  2. Line 230: What size diamond and Sic abrasive particles were used in the experiment, this is necessary information to interpret the test results.
  3. Line 255: What kind of three-dimensional (3D) laser microscope was used in the experiment, what magnification, etc.?
  4. Please use the parameters for assessing surface roughness throughout the entire section 4.1 Effect of compound magnetic field strength on roughness. This is not precise information "initial roughness values in the range of 1.06-1.1µm".
  5. Line 267: It is not clear from the content what kind of abrasive particles were used, whether diamond or SiC.
  6. In section 4.2 Effect of rotation speed on roughness It is not clear from the content what kind of abrasive particles were used, whether diamond or SiC.
  7. In subsection 4.3 Effect of type of abrasive particle on roughness, information about the size of the abrasive particles used is necessary, only with this information we can compare the results of the experiment.
  8. For sure, the article would be richer if you show sample images of the geometric structure of polished surfaces in 3D, because the measuring device used allows it.

Reviewer 3 Report

The review concerns the work entitled: “Annular surface micromachining of titanium tubes using a magnetorheological polishing technique”.
The paper includes the following sections: Introduction, Design of MR polishing apparatus, Experimentation, Results and Discussion, Conclusions.

This paper requires major improvements to be made according to the suggestions enlisted below.

1. The nomenclature should be systematized. Abbreviations and symbols should be expanded (explained) in the place where they appear for the first time in the text, example - MR. This abbreviation is explained more than two times (Abstract, Keywords, Introduction, …). I suggest enumerating necessary abbreviations and symbols used in the text at the beginning of the paper.

2. The Authors try to analysis the surface roughness (paragraph 4), but they are not writing about it in the abstract as well as in keywords.
3. Page 5. Indexing. Small or large letter? – it means ‘P’ or ‘p’ as an index for A and d?

4. In the text, the Authors used the terms: ‘surface roughness’ or ‘roughness’ or ‘roughness value’. What did the Authors mean when using these sentences?

5. Paragraph 3.
In the manuscript, there are not detail about surface roughness measurements. On page 8 the Authors wrote: “A three-dimensional (3D) laser microscope and electronic analytical balance were used to measure the surface roughness and specimen mass, respectively. In order to minimize the man-made operation error and machine error, three points at different locations were selected as the measuring points to measure the surface roughness. Three repeated measurements at each point were taken to obtain the average value of these three data as the experimental data.” It is too poor information (!). The reader should know, what was the method of surface measurement, size of measurement area - these are not "three points at different locations" as the Authors wrote (!), method of form removal (if the samples are not flat, the parameters of surface texture should be generated after form removed), and what kind of surface texture parameters were analysed (probably the Authors were analysed only one chosen parameter, but I do not know which and why exactly this parameter). There is no information about the kind of surface texture parameter which was analysed.

6. Paragraph 4.
6.1. There is no information about the kind of surface roughness parameter which was analysed (see comment 5).
6.2. In all of the figures are missing the kind of surface texture parameter. The Authors present results for single surface measurement (not for three surface measurements, as they declared in paragraph 3.3).
6.3. Moreover, I do not see the results of surface roughness measurements as surface views. I expect the surface images as an isometric view or pseudo-colour image. The Authors only wrote about some values as results of roughness measurement.
6.4. Page 10, Figure 11. In the manuscript the Authors present results from SEM, but they did not write about detail (device, method of study, etc.)

7. Conclusions should be correct and supplemented, taking into account the previous comments.

Round 2

Reviewer 1 Report

Authors have addressed all the comments satisfactorily.

Reviewer 3 Report

Dear Authors,
Generally, I accept the revised version of the manuscript.
I suggest use italics for all physical symbols (including Ra parameter) as well as correct the quality of Figure 9.

Best regards.